# Impact of Plasma Combustion Technology on Micro Gas Turbines Using Biodiesel Fuels

**Ahmed M. R. N. Alrashidi \*, Nor Mariah Adam, Mohd Khairol Anuar Bin Mohd Ariffin** , **Alajmi Fnyees, Amer Alajmi, Alrashidi Naser and Hairuddin Abdul Aziz**

Department of Mechanical and Manufacturing Engineering, Faculty of Engineering, Universiti Putra Malaysia (UPM), Serdang 43400, Malaysia; mariah@upm.edu.my (N.M.A.); khairol@upm.edu.my (M.K.A.B.M.A.); alktab@hotmail.com (A.F.); amerq880@gmail.com (A.A.); nasser.albarak@gmail.com (A.N.); ahziz@upm.edu.my (H.A.A.)
\* Correspondence: ahmadmrn@yahoo.com; Tel.: +96-55-0600-607

**Abstract:** The adoption of biorenewable alternative fuel resources from biofuels (ethanol or biodiesel) has produced promising solutions to reduce some toxic greenhouse gas (GHG) emissions from gas turbine engines (GTEs). Despite the reduced hydrocarbon associated with adopting alternative bio-renewable fuel resources, GTE operations still emit toxic gases due to inefficient engine performance. In this study, we assess the impact of the integration of plasma combustion technology on a micro-GTE using biodiesel fuel from animal fat with the aim of addressing performance, fuel consumption, and GHG emission reduction limitations. Laboratory design, fabrication, assembly, testing, and results evaluation were conducted at Kuwait's Public Authority for Applied Education and Training. The result indicates the lowest toxic emissions of sulfur, nitrogen oxide (NO), $NO_2$, and CO were from the biodiesel blended fuels. The improved thermal efficiency of GTE biodiesel due to the volume of hydrogen plasma injected improves the engine's overall combustion efficiency. Hence, this increases the compressor inlet and outlet firing temperature by 13.3 °C and 6.1 °C, respectively. The Plasma technology produced a thrust increment of 0.2 kgf for the highest loading condition, which significantly impacted horsepower and GTE engine efficiency and reduced the cost of fuel consumption.

**Keywords:** plasma technology; biodiesel; gas turbine; greenhouse gas; rich fuel; fuel atomization



## 1. Introduction

The use of fossil fuel technology in the operation of gas turbine engines (GTE) faces issues that include low thermal efficiency, poor atomization, low vapor pressure, and high greenhouse gas (GHG) emissions [1,2]. Thus, our research motivation was to restructure the design principle of gas turbines to enhance performance and reduce fuel consumption and GHG emissions [3]. The use of biofuel as an alternative energy source in transportation systems has gained attention and interest as an alternative for automobiles in developed nations [4–6]. The environmental benefits motivated this study, considering the low greenhouse gas (GHG) emission content of sulfur and hydrocarbons in bio-renewable fuel resources, as well as the associated improved engine performance [7–9]. In particular, the benefits of biodiesel as an alternative aviation fuel have attracted research interest in addressing the inefficient performance of conventional fossil fuels [10]. The incomplete combustion of fossil fuel hydrocarbons adversely impacts GTE performance in terms of the GHG emission levels, fuel consumption, engine running cost [11–13].

A typical gas turbine is characterized by a continuous-flow engine and steady flame production during the combustion process, with high hydrocarbon emissions from conventional fossil fuel resources [14,15]. However, these emissions may negatively impact the environment by depleting the ozone layer. The gas-turbine architecture permits various fuels that support complete combustion in the engine. Some GTE features, such as

moderate compression ratios, robust mechanical designs, and versatile combustion systems enhance the engine's potential to utilize a wide variety of biofuels (such as alcohols, biodiesel, low-calorific-value (LCV) gasified biomass, synthetic gas, hydrogen [16], and natural gas) [15,17]. It is worth mentioning that fuel properties influence the performance efficiency of gas turbines and also determine the final composition of emitted greenhouse gases (GHG), such as nitrogen oxide ($NO_2$) and carbon monoxide (CO) [18–20]. Efforts have been made to apply biodiesel fuel sources to power GTEs for improved performance efficiency and reduced GHG emissions [18,21]. High emissions of sulfur oxide (SO), CO, and NO are known to result from the combustion of conventional fossil fuels in typical GTE systems [21,22]. Factors such as injection timing, adiabatic flame temperature, radiation heat transfer, and injection delay are also responsible for higher $NO_2$ emissions in reciprocating engines [23,24]. The antioxidant additives in biodiesel mainly contain phenolic groups, which are more likely to form soot than glyceride impurities, even with the potential to burn more cleanly than fossil fuels [25–27]. Control of emission levels remains an challenge; there is a need for more sustainable, cheaper, and environmentally friendly alternative energy sources for GTE aerospace applications.

Animal fats are a promising alternative for biodiesel production but require more complex processing than natural oils. In Kuwait, fat waste from sheep is the primary feedstock for biodiesel production with high contents of free fatty acids (49.1 mgKOH/g). Fat waste is sustainable, abundant, and inexpensive and reduces handling and risk of impacting on the environment without economic competition, as seen in vegetable oils [28]. In addition, use of jet fuel in the operation of engines has the disadvantages low thermal efficiency, poor atomization or vitalization, and low vapor pressure. These factors, if not addressed, result rapid evaporation and high operation cost, leading to increased fuel consumption [29]. To overcome these limitations, effective deployment of biodiesel fuel in GTE operation is among the best options to reduce GHG emission levels. Plasma technology requires a higher temperature for effective biodiesel combustion and enhanced performance in a gas turbine engine.

Plasma technology is a solution based on free-electron formation under high temperatures to enhance combustion efficiency in automotive engines with biofuels [30–32]. Plasma technology has evolved over the years but has not been applied to GTE. Current plasma designs can easily reach very temperatures of over 5000 °C, making them suitable for used in GTEs [33,34]. Therefore, a fundamental understanding of plasma–fuel combination and its correlation with emission levels is essential for optimal use of biofuel in GTEs. Considerable progress has been made in recent research in terms of understanding the impact of plasma on improving the fuel combustion process [35–37]. The validation of such mechanisms was achieved through experimentation under controlled conditions and by comparing the results with numerical simulations of discharge and combustion processes [38,39]. There is no detailed review of the recent applications of plasma in internal combustion engines, particularly in GTE applications. This knowledge gap is a severe setback in advancing the science of plasma technology in internal combustion (IC) engines. The use of biodiesel from plant-based oils results in competition in food supply chains, but biodiesel from animal fat waste is cost-effective and reduces environmental impact. Therefore, there is an opportunity to integrate plasma technology and biodiesel from cost-effective Kuwaiti sheep fat waste in GTEs. We investigated the feasibility of injecting a hybrid plasma-rich animal fat biodiesel mixture in the compressor inlet to enhance the efficiency of GTE operation and reduce GHG emission levels. The integration of plasma technology in GTEs is proposed as a solution based on the principle of free electron formation under high temperatures to enhance the overall combustion efficiency with the adoption of biofuels, meeting the higher combustion temperature requirement for the oxygenated biodiesel fuel [40]. A fundamental understanding of the role of plasma–fuel combination in reducing emission levels can help to achieve optimal use of biofuel [41].

There is a need to restructure the design principle of significant aspects of GTEs to improve engine performance and reduce fuel consumption, cost, and GHG emissions,

which motivated the current study. The main objective of this study is to investigate and assess the impact of plasma combustion technology on a mini gas turbine (MGT) using animal fat biodiesel fuel. This is achieved by injecting a hybrid plasma-rich mixture in the compressor inlet of the engine system, achieving enhanced combustion of biodiesel fuel through improved thermal efficiency. This study addresses the knowledge gap in the application of plasma technology and animal fat biodiesel fuel in IC engines. Such applications could reduce GHG emissions, improve engine performance efficiency, and reduce fuel consumption associated with GTE operation. The remainder of this paper is structured as follow. In Section 2, introduce the research methods entailing fuel characterization, laboratory mini GTE fabrication, construction, assembly, and testing. Data collection, analysis, and discussion are detailed in the following section. The final section is the conclusion and recommendations for future work.

## 2. Materials and Methods

This research focused on studying plasma combustion technology for MGT engines using biodiesel fuel resources. We focused on the external integration of a hybrid plasma-rich fuel mixture in the combustion chamber intake manifold of the proposed fabricated 50 kW (67 hp) testbed MGT engine. We used several biofuel mixtures for the operation of the MGT engine under three different loading conditions. We fabricated the MGT engine system in our laboratory using individual engine components, such as compressor systems, a plasma torch unit, and GTE system assembly modifications. Other operations involved fuel system design, air intake system construction, and atomization construction for the testbed engine. We investigated the viability of adopting plasma technology and alternative renewable biodiesel fuel (from animal fat waste) in MGT engines. The findings should be useful in enhancing effective engine performance and reducing operating costs and GHG emissions.

### 2.1. Fuel Characterization

We adopted biodiesel fuel extracted from animal fat organic waste (B20, B50, B75 and B100) blended mixtures with conventional diesel, as well as kerosene (control) aviation fuel for the operation of the fabricated MGT engine. The six proposed six liquid fuels require stringent preconditioning before adoption in GTEs due to their more comprehensive range of hydrocarbons and particulate contents. The physical and chemical properties of each proposed fuel for the operation of the MGT engine were characterized to investigate the engine's combustions performance [42]. The physical and chemical properties of the fuel must be determined for significant adoption GTE operations [43]. The characterization of these is based on selected factors (specific gravity, density, kinematic viscosity, total acid number, water content, total sulfur, flash point, lubricity, cloud points, and pour points). These were validated in compliance with the American Society for Testing and Materials (ASTM) D6751 standards [44]. The test was conducted at the Petroleum Research Center, Kuwait Institute for Scientific Research. The properties of viscosity, density, and surface tension play a central role in spray combustion. Fuel viscosity affects the power required to pump the fuel through the fuel system, as well as atomization and droplet evaporation. The higher the viscosity of the fuel, the lower the quality of atomization, leading to soot formation and resulting in carbon deposits within the combustion system. Carbon deposits can damage hardware due to high thermal radiation and clogging. Most liquid fuel systems also require separate air atomizing systems during the ignition process, depending on the type of atomizer used in the fuel injection system. The six characterized fuel samples for fabricated MGT testbed engine deployment are displayed in Figure 1.

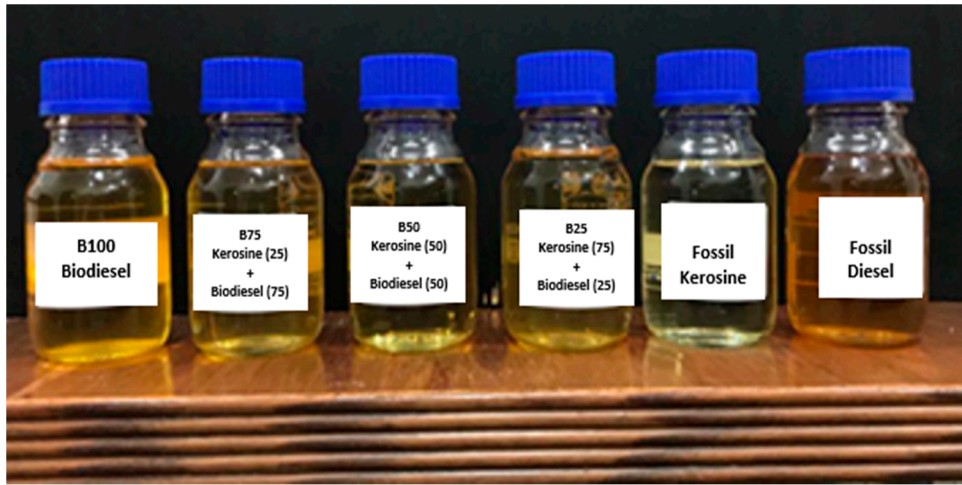

**Figure 1.** Characterized liquid fuel samples.

### 2.2. Micro Gas Turbine (MGT) Engine Fabrication

Several studies have been executed to improve the fossil fuel combustion processes in GTEs. The motivation of the current study was triggered by the need to restructure the design principle of some significant aspects of the engine to enhance performance, reduce fuel consumption cost, and reduce GHG emissions. We aimed to fabricate a micro-GTE in a laboratory setting with the external modification of the compressor intake with an integrated plasma torch and fuel atomizer devices. The proposed approach represents a novel external integration of a hybrid plasma-rich fuel mixture by injection in the compressor inlet of an MGT engine.

With the integration of plasma technology, we aim to produce a hydrogen-rich plasma to aid a smooth combustion process through the partial oxidation of 6 different liquid fuels from kerosene, diesel, and extracted animal fat biodiesel blended mixtures (B20, B50, B75, and B100). A schematic diagram of the integrated scheme redesign is displayed in Figure 2. The GTE nomenclature is shown in Table 1.

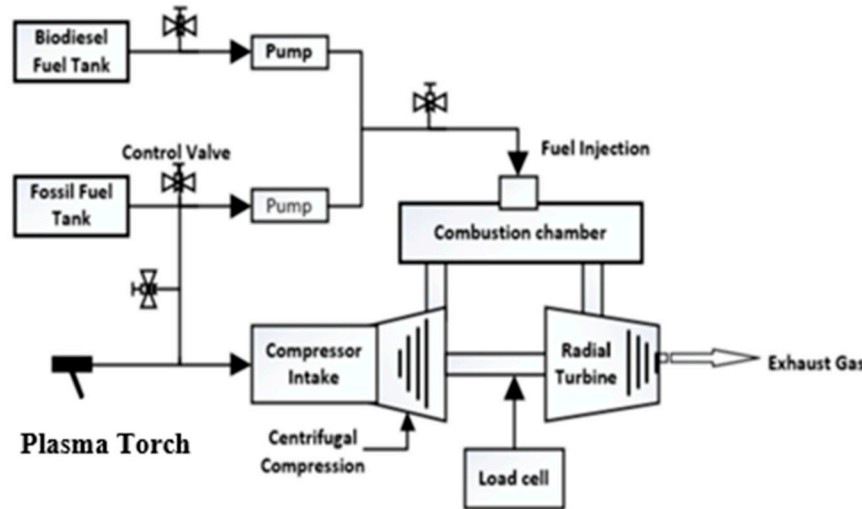

**Figure 2.** Schematic diagram of the fabricated micro gas turbine engine.

**Table 1.** Micro gas turbine nomenclature.

| S/n | Component | Specifications |
|---|---|---|
| 1 | Power | 50 kW (67 hp) |
| 2 | Mechanical efficiency | 78% |
| 3 | Compression flow rate | 58 ips/m (26.37 kg/s) |
| 4 | Maximum pressure ratio | 2.6 |
| 5 | Plasma torch temperature | 5000–8000 °C |
| 6 | Fabricated fuel tank storage capacity | 20 L in volume |
| 7 | Fuel pump pressure | 8-bar |
| 8 | Fabricated and assembled starter system | 300 L cylindrical air tank |
| 9 | Fabricated ignition system | 240 V to 40 kV transformer and a spark plug |

GTE fabrication, assembly, testing, and evaluation were conducted at Kuwait's Public Authority for Applied Education and Training. This is the largest academic institution in the Middle East with well-equipped for GTE fabrication by deploying a turbo-compressor, combustion chamber, plasma torch, atmospheric air, fuel source, and ultrasonic atomizer salvaged from an existing engine. The following procedures were followed in for fabrication of GTE with an external integrated hybrid plasma-rich fuel atomizer at the compressor inlet of the system.

### 2.2.1. Turbocharger Selection

The pressure ratio and airflow rate were used to estimate the compressor speed (rpm) and required engine efficiency. This should be within an acceptable operation range (generally between the surges and choke lines), as displayed in Figure 3a. A turbocharger was selected based on a maximum mechanical efficiency of 78%, a maximum pressure ratio of 2.6, and a compression flow rate of 58 ips/m (26.37 kg/s). Based on the selected turbocharger parameters, an equivalent combustion chamber was salvaged from an existing jet engine to adapt to the fabricated testbed engine, as displayed in Figure 3b. The fuel combustion takes place in a chamber fitted with four injectors to introduce and atomize the charged lean fuel mixture into the combustion chamber. The combustion chamber is linked directly with the turbine wheel to prevent energy loss. The turbine wheel is radial in shape with an inlet port for the combustible gases and an exit port for the GHG exhaust emissions.

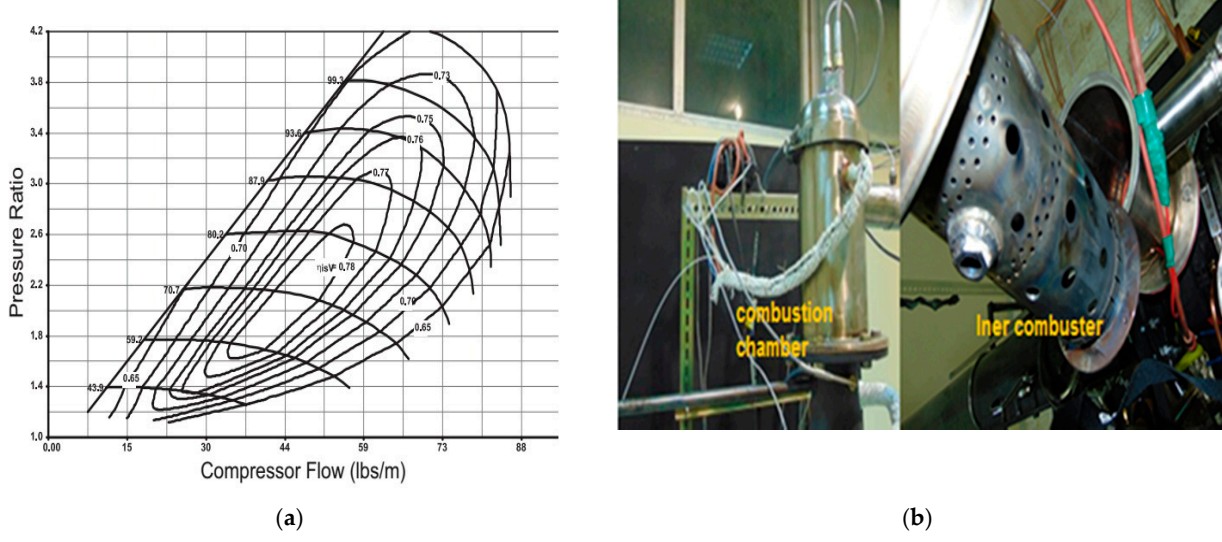

(**a**)                                                     (**b**)

**Figure 3.** Fabricated GTE (**a**) compressor map and (**b**) adapted combustion chamber.

### 2.2.2. Redesigned Exhaust Nozzles

Four injector nozzles salvaged from an existing jet engine installed the engine's inlet manifold. We adapted the nozzles to the fabricated MGT engine to enhance measurement of exhaust gas parameters and other relevant performance information, such as engine thrust and GHG emissions, under varying engine loading conditions with all proposed fuels.

### 2.2.3. Fabrication of Oil and Fuel System Unit

The fabricated parts of the oil system for the MGT engine consist of a 14-L oil tank, a gear oil pump, and a heat exchanger. The fabricated oil system unit comprises an oil filter and a control valve to maintain the system oil pressure at 5 bars for sufficient lubrication of all moving parts. It also serves as a cooling medium, assisting in quick heat dissipation to protect the turbine engine from excessive heating, as displayed in Figure 4a. The oil used in the system is fully synthetic of 5 W-30 grade, in compliance with the Petroleum Quality Institute of America as specified for the smooth operation of turbochargers of any GTE.

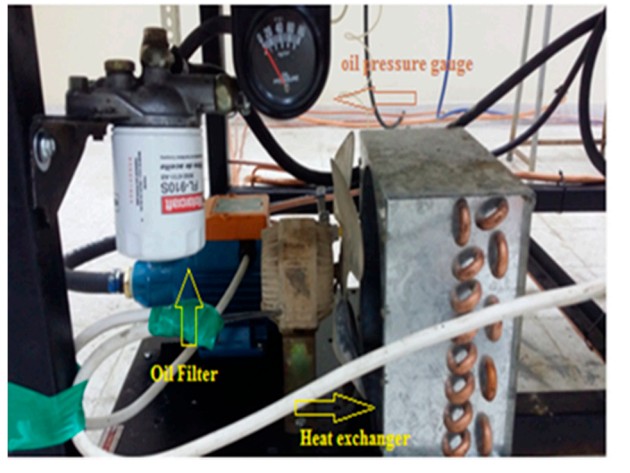 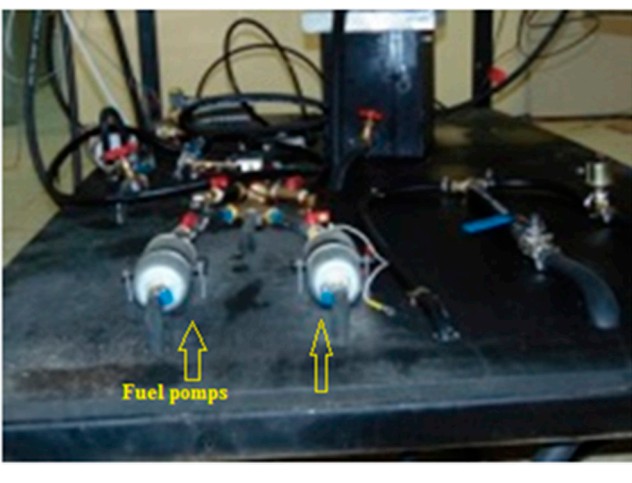

(**a**) (**b**)

**Figure 4.** Fabricated GTE (**a**) oil system unit and (**b**) fuel system unit.

The fuel system consists of two distinct subsystems: one for the fossil kerosene and the other for the biodiesel fuel system, as displayed in Figure 4b. The storage capacity of the fabricated tanks is 20 L in volume, with an 8 bar lifting pressure fuel pump (SKU 456072355_MY-702440838) for each fuel line and a pressure-controlled valve and injector nozzle. The system fuel line has an integrated fuel filter to prevent particle blockage of the injector's nozzles.

### 2.2.4. Starter and Ignition System Unit Fabrication

The fabricated and assembled starter system for the proposed MGT engine comprises a 300-L cylindrical air tank connected to a compressor for the effective delivery of dry air, with a connected solenoid valve for air control. A nozzle was attached to the compressor to eject the stream of air directly onto the fan blades of the compressor, initiating rotation. The momentary impact of the compressed air on the edges results in a high-speed rotation speed of 13,000 rpm as an initial startup for the turbine system from the resting position. Figure 5a displays a picture of the fabricated starter units adopted for the initial startup of the MGT engine testbed.

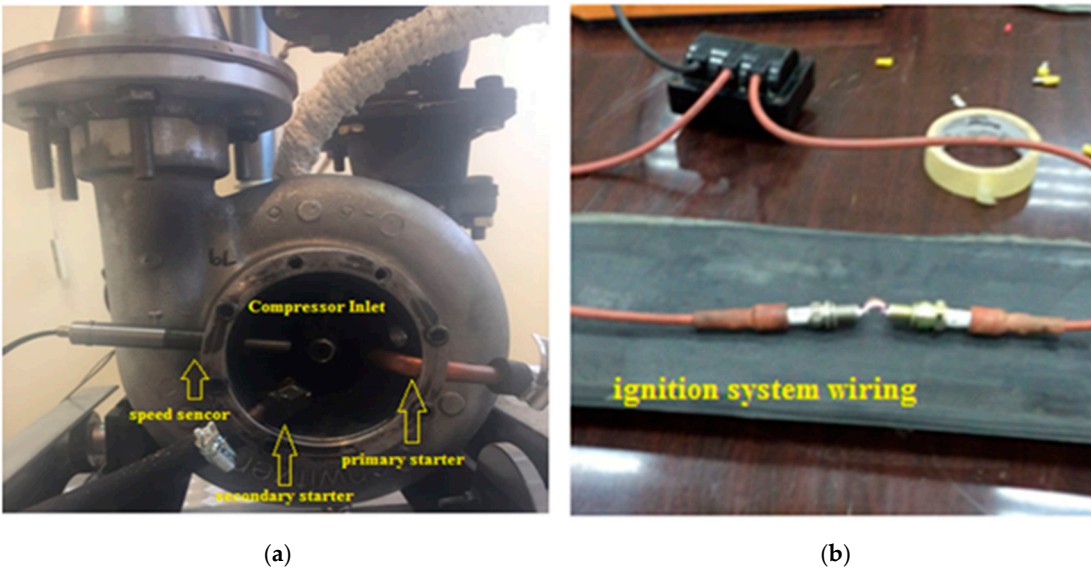

|       |       |
| :---: | :---: |
| (**a**) | (**b**) |

**Figure 5.** Fabricated GTE (**a**) starter unit and (**b**) ignition system.

The fabricated ignition system consists of a 240 V to 40 kV wire coil step-up transformer and a spark plug. The unit enables the spark plug to conduct an electrical discharge through the air gap between the poles to generate an arc between the plug heads, initiating the required spark for the combustion system within the combustion chamber of the MGT engine, as depicted in Figure 5b. The resultant electrical spark is converted into heat, which eventually ignites the fuel mixture atomizer in the ignition chamber.

### 2.2.5. Air Intake System

The need to fabricate an air intake system as an oxygen source is an essential requirement to aid in fire ignition and support effective continuous combustion and the continuous burning process for the MGT engine system. The major components of the fabricated air intake system include two suction apertures with separate air filters and motorized throttle valves for control of atmospheric air intake. The motorized throttle valve, as well as each intake, is fitted with a 5-inch filter to control the mass and pressure of the air sucked through the apertures. Both intake apertures meet on an 8-inch diameter header connecting pipe at the main junction, as displayed in Figure 6a.

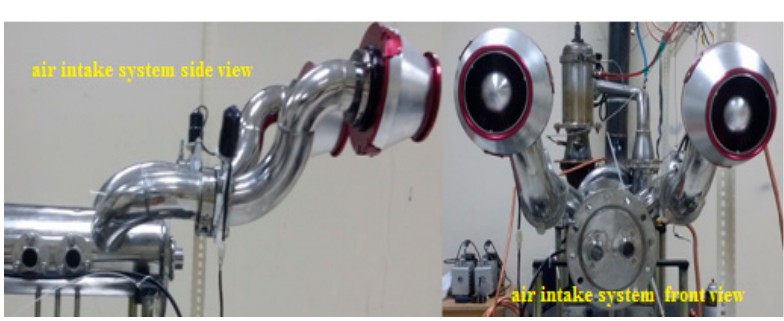

(a)

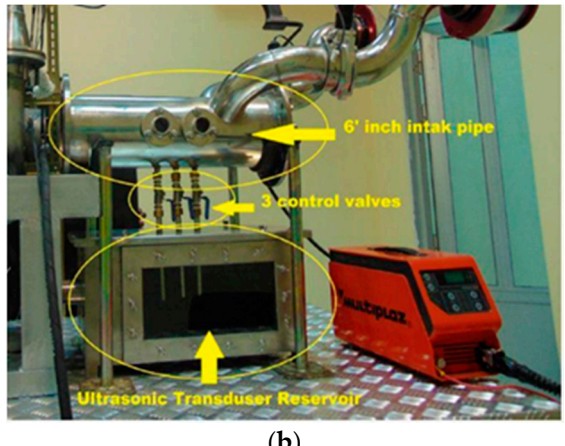

(b)

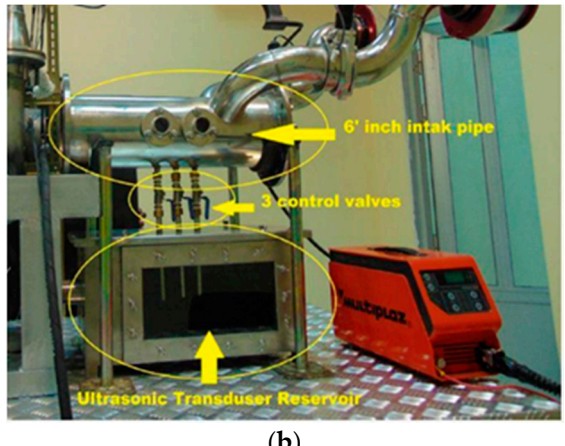

(c)

**Figure 6.** (**a**) Fabricated GTE air intake/ exhaust system. (**b**) Fabricated GTE ultrasonic atomizer. (**c**) Experimental setup of Plasmatron integration on MGT engine.

2.2.6. Fuel Ultrasonic Atomizing System

We construction and fabricated this unit as an external integral part of the MGT engine testbed due to the need to generate pure hydrogen vapor. This hydrogen is exposed to the integrated Plasmatron device at the compressor's intake to generate the required hydrogen plasma to be compressed to aid in further combustion in the combustion chamber. This process is called atomizing, whereby the necessary hydrogen-rich fuel mixture must be created separately under different reaction conditions through a process known as partial oxidation (POX), as earlier adopted in an experiment in an internal combustion engine coupled to a plasma fuel reformer [40].

In this study, we proposed an external integration of a hybrid plasma-rich fuel mixture into the intake manifold of the MGT engine to produce a fine fog of the rich fuel mixture from a fossil kerosene tank, external air, as well as an ultrasound device. The sonification effect resulted in fine fog from the fuel- and air-rich mixture, which supplies the plasma needed to generate hydrogen gas, as seen in Figure 6b. The atomizing system works directly with the air intake system. The air intake system provides a suitable environment to pull the fuel fog from the ultrasound device into the plasma device through the compressor to the combustion chamber of the MGT engine.

### 2.2.7. Plasma Unit Selection and External Integration

The choice of plasma device for this study was influenced by factors such as the required temperature for the plasma torch, the cost implication of different plasma torches on the market, and applications in previous studies. The main advantage of a plasma-based igniter compared to a conventional spark plug is the considerably higher generated plasma plume volume and velocity, as seen in the adopted Plasmatron torch, with a generated temperature range between 5000 and 8000 °C, allowing for deeper penetration of a highly reactive plasma plume into the combustion zone of an engine, as proposed in [45].

The developed testbed produced plasma from water vapor at a pressure of 1 bar (14.7 psi). The device consists of a 240 volt, 3.5 kW electrical source power unit; the plasma torch was positioned in the front of the fabricated compressor inlet (after many trials) [46] for hydrogen gas reformation. The hydrogen gas reformed at 5000–8000 °C is transmitted into the combustion chamber, as seen in Figure 6c.

The reformed gas is injected into the combustion chamber, and the compressor system increases the combustion efficiency, as previously reported [36]. The Plasmatron devices used in this study also provide ohmic heating of gases to an elevated temperature at which the gas is partially ionized as a requirement to aid in heat conduction. A wide range of applications is possible, from oxidation to steam reforming, boosting the reaction rate that occurred through the creation of a small region of very high temperature from 5000 to 8000 °C. In this situation, radicals are produced, increasing the average temperature in the surrounding area. Adequate safety precautions were taken in handling the Plasmatron system, including placing the MGT engine testbed in a smaller room inside the laboratory with double thickness and a view window. This will support the complete combustion of the proposed fuels to limit GHG emissions in gas turbine engines.

### 2.2.8. Control and Measuring Unit

The control system was designed for the effective operation of the fabricated micro-GTE. The unit consists of several pipes welded at different levels, plumbing lines for fuel, electrical wiring work, and sealing to prevent leakage of fuel, liquid, gases, and produced plasma flow. All computers and electronics were placed outside the room; exhaust gas was vented vertically 4 m into the atmosphere. There were no adjacent buildings near the stand-alone laboratory, and fire extinguishers were located outside and inside the test rooms. Appropriate PPE (goggles, gloves, face shields, aprons/lab coats, and safety boots) were adorned while working in the laboratory.

### 2.3. Fabricated Micro Gas Turbine Assembly

The assembly of all fabricated system units complied with all safety rules and operational standards as stipulated [47,48]. The entire assembled GTE with the proposed external integrated electrically powered Plasmatron torch and ultrasonic atomizer for hydrogen gas reformation is displayed in Figure 7. The plasma technology integration approach reduces toxic GHG emissions and improves system performance impact on the fuel consumption rate, thereby reducing the system operating cost across several fuel types. The details of all parts are listed in Table 2.

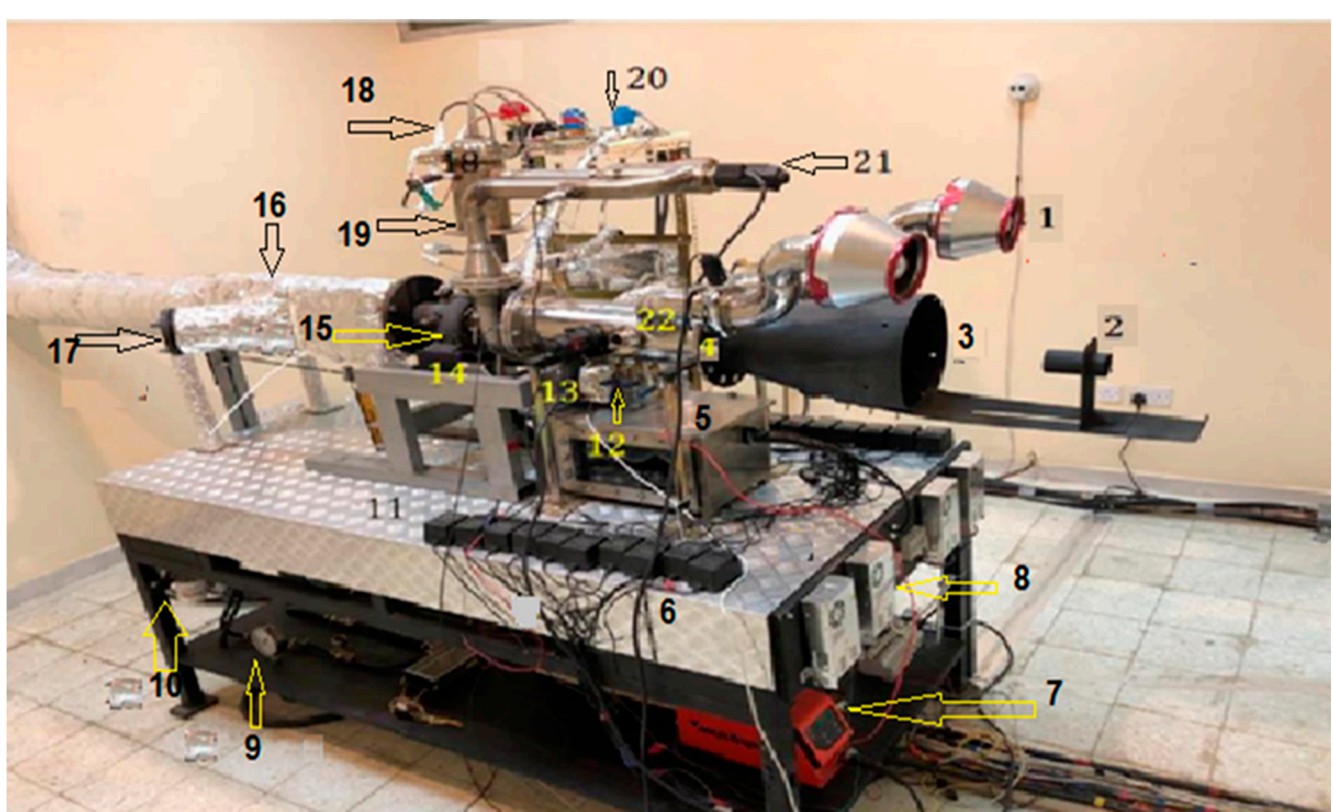

**Figure 7.** Complete assembly of fabricated GTE.

**Table 2.** Gas turbine engine parts.

| Part Number | Description | Part Number | Description |
|:---:|:---:|:---:|:---:|
| 1 | Intake pipe | 12 | Load cell |
| 2 | Atomisation point | 13 | Plasmatron torch |
| 3 | Intake main pipe | 14 | Engine core |
| 4 | Intake nozzle | 15 | Turbocharger |
| 5 | Ultrasonic tank, | 16 | Exhaust pipe |
| 6 | Ultrasonic transformer | 17 | Exhaust sight glass |
| 7 | Plasmatron unit | 18 | Electric igniter |
| 8 | Ultrasonic transformer | 19 | Combustion chamber |
| 9 | Oil pressure gauge | 20 | Fuel solenoid valves |
| 10 | Oil filter | 21 | Plasmatron torch |
| 11 | Engine base | 22 | Engine intake tunnel |

*2.4. Micro GTE Operation, Testing, and Measurements*

Initially, the turbine engine was operated to attain a stabilized temperature for about 20 min. After stability, three different fuels obtained from conventional fossil diesel, kerosene, and biodiesel were used to operate the fabricated MGT engine. Six different fuels were adopted to operate the fabricated MGT engine under normal operation before the external integration of the hybrid plasma-rich fuel mixture to measure the fabricated GTE's performance and GHG emission levels.

Similarly, after injection of the hybrid plasma-rich fuel mixture (Figure 7), the system was operated at a fixed air-to-fuel ratio of 2:1 and a varied thrust load of 10, 20, and 30 psi in both operational scenarios with and without external integration of the proposed

hybrid plasma-rich fuel mixture. The complete fabricate MGT setup was subjected to three different loading conditions (low load, medium load, and maximum loading) and six fuel types (use fossil kerosene; fossil diesel; Kuwaiti sheep fat biodiesel fuel, and different blend ratios between renewable biodiesel fuels and kerosene) and operated under two different operational scenarios:

i.    Normal operation of the mini-GTE without integrated plasma-rich fuel mixture;
ii.   External integrated plasma-rich fuel mixture in mini-GTE.

Operational testing, data acquisition, measurement, and analysis of the wholly assembled micro-GTE were executed in the laboratory setup to determine effective engine performance based on fuel consumption rate and reduction and control of toxic gas emissions. Several parameters, such as GTE combustion temperature, compressor pressure, harmful GHG emission qualities, engine fuel consumption rate, engine performance, and loading thrust, were monitored using different sensors at different locations for future analysis.

## 3. Results and Discussion

### 3.1. Fuel Characterizations Results

The characterization result from the six different fuel samples used in the operation of the fabricated GTE are in line with the American Society for Testing and Materials (ASTM) D6751 standards [49], as presented in Table 3. The physicochemical characteristics results indicate that the pure biodiesel fuel (B100) from animal fat has the highest kinetic viscosity of 4.339 cSt, a lubricity value of 547.5 μm, and oxidative stability compared with the obtained result from other biodiesel blends, fossil diesel, and conventional fossil kerosene fuel. The sulfur content is the lowest across all biodiesel blends compared with fossil diesel and kerosene, with high hydrocarbon and sulfur impurity contents. The sulfur poisonous substance must be eliminated to reduce the $SO_2$ and $NO_2$ emissions into the atmosphere, and this prevents blockage of the engine catalytic converters that help minimize particulate emissions. This result implies that the blended biodiesel mixtures extracted from animal fats have potential applications as biorenewable organic fuel to power diesel engines and GTEs, considering the environmental benefits, including a reduction in toxic gases.

**Table 3.** Properties of selected fuels.

| Sample ID | Fossil Fuel | | Biodiesel Fuel | | | |
|---|---|---|---|---|---|---|
| | **Diesel** | **Kerosene** | **B20** | **B50** | **B75** | **B100** |
| Specific gravity | 0.8418 | 0.7906 | 0.8088 | 0.8299 | 0.8555 | 0.8728 |
| Density at 15 °C (g/cm$^3$) | 0.8410 | 0.7898 | 0.80796 | 0.8291 | 0.8546 | 0.8719 |
| Density at 25 °C (g/cm$^3$) | 0.8339 | 0.78223 | 0.8006 | 0.8219 | 0.8576 | 0.8649 |
| Kinematic viscosity 40 °C (cSt) | 3.5819 | 1.2144 | 1.5832 | 2.1677 | 3.3284 | 4.339 |
| Total acid number (mgKOH/gm) | 0.0110 | 0.01244 | 0.0321 | 0.0570 | 0.1778 | 0.1035 |
| Water content (KF) (ppm) | 83.28 | 87.48 | 251.1 | 419.4 | 134.0 | 790.2 |
| Sulfur content (mg/L) | 620.17 | 92.17 | 73.98 | 55.02 | 30.07 | 9.06 |
| CFPP (°C) | −5.0 | <−51.0 | - | −6.0 | 1.0 | - |
| Flash point | 90 | 47 | - | 89 | 62 | 97 |
| Lubricity at 60 °C (μm) | 528 | 524 | 466.5 | 457.5 | 508 | 547.5 |
| Pour point (°C) | 0 | −54 | −12 | 0 | 6 | 15 |
| Cloud point (°C) | 0 | −53.8 | −3.5 | 6.5 | 6.9 | 11.2 |
| Ca (mg/kg) | 0.00 | 0.00 | 0.08 | 0.28 | 0.06 | 0.38 |
| Mg (mg/kg) | 0.02 | 0.02 | - | 0.06 | 0.02 | 0.10 |

In the Table 3, green and yellow highlights indicate the improved biodiesel parameters from the characterization test.

In contrast, the high viscosity, volume flow rate, and spray characteristics could adversely affect the injection system at the engine's manifold unit if not enhanced. Therefore,

the blended biodiesel mixtures require modification to improve compatibility with low kinematic viscosity for effective combustion at much lower heating temperatures. It can be observed that the flashpoint value for B75 blended fuel showed the closest recorded flash point of to that from the fossil kerosene, at 62 °C. B75 also demonstrated lower sulfur content in GHG emissions within the acceptable range for optimal GTE operation.

### 3.2. Gas Turbine Compressor Inlet/Outlet Temperature

The temperature of the assembled unit was taken using several 2 m length specialized thermometer instruments (model no: MT-TCE 50 and Part No: MT-TCE 50 K) installed strategically on the fabricated MGT engine and other auxiliary devices to monitor several temperature readings, including the surrounding ambient temperature, the temperature before the compression process, the temperature after the compression process, the exhaust temperature, and the engine oil-cooled kerosene and biodiesel fuel temperatures.

The impact of the external integrated plasma technology on the compressor inlet temperature is reflected in the 10 °C increase recorded across all applied fuels compared with GTE operation under normal conditions, as highlighted in Table 3. The result supports the need for increased temperature of the combustion chamber for the effective performance of the engine. A slight increase in firing temperature significantly impacted the produced horsepower and engine efficiency. Continuous heating is achieved with plasma technology compared with regular GTE operation without plasma technology under the three different loading conditions in Table 4. The increased inlet temperature supports the combustion chamber temperature buildup for effective combustion performance of the engine in order to achieve continued heating. The B20 biofuel blend recorded the highest inlet temperature under low and medium loading, with an increase in loading conditions across all biodiesel blended fuels. Under the external integrated plasma technology, B75 and B100 recorded the highest inlet temperatures at the most elevated loading conditions (Figure 8a). This may be due to the high-temperature characteristics of plasma technology, which influence the spectacular increase in the load. The impact of load increase affected the change in temperature measured at the compressor inlet, as observed for all fuel temperature measurements. The higher the loading, the greater the compressor inlet temperature.

**Table 4.** Compressor inlet temperature under different loading conditions.

| Mini GTE Loading | GTE Operation Scenarios | Compressor Inlet Temperature Across Fuel Samples (°C) | | | | | |
|---|---|---|---|---|---|---|---|
| | | Diesel | Kerosene | B20 | B50 | B75 | B100 |
| Low | Normal Operation | 31.9 | 30.1 | 28.9 | 28 | 28.9 | 31.6 |
| | Plasma + Rich Mixture | 32.5 | 32.1 | 39.8 | 32.8 | 31.9 | 35.4 |
| Medium | Normal Operation | 32.5 | 30.7 | 30.7 | 30.5 | 30.5 | 28.6 |
| | Plasma + Rich Mixture | 37.9 | 30 | 40.1 | 39.4 | 37.4 | 39.3 |
| High | Normal Operation | 32.3 | 29 | 31 | 31 | 31.6 | 33.1 |
| | Plasma + Rich Mixture | 43.9 | 41.6 | 35.4 | 41.9 | 44.9 | 45.5 |

In the Table 4, yellow highlights indicate the biodiesel improved value from integrated plasma and rich fuel mixture.

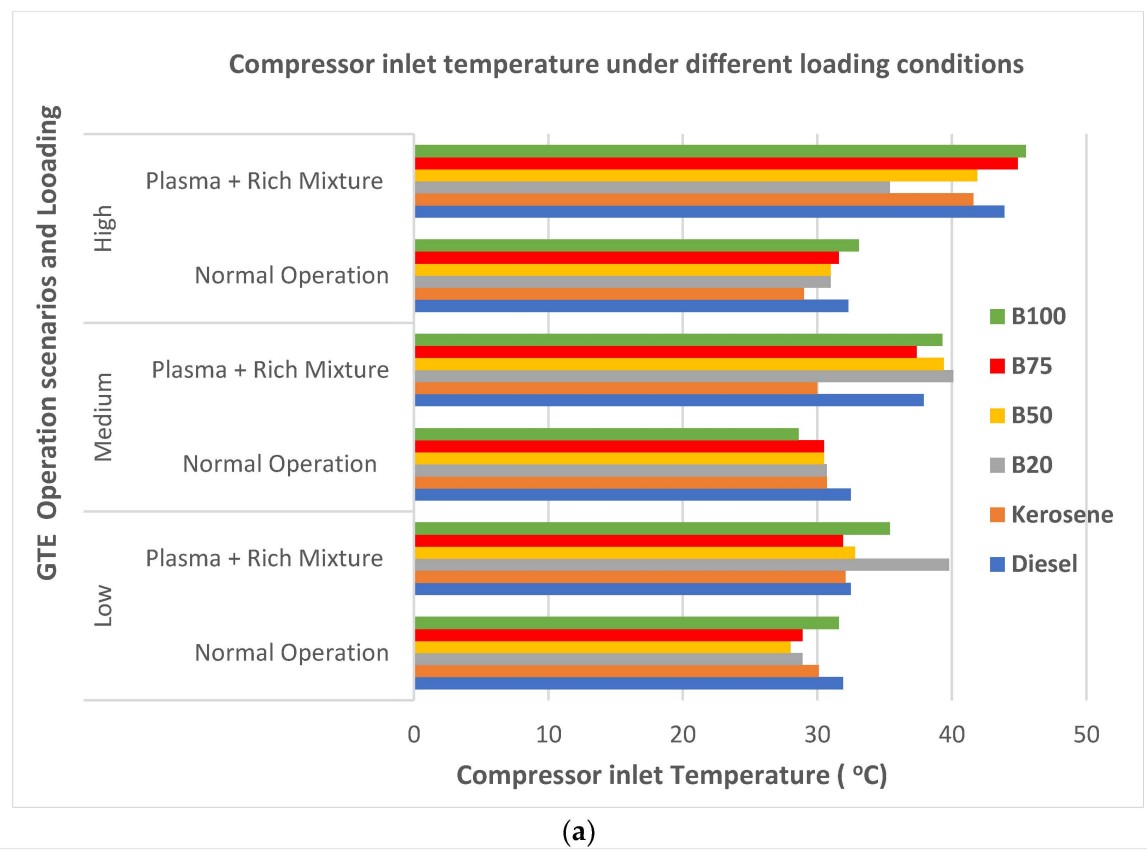

(**a**)

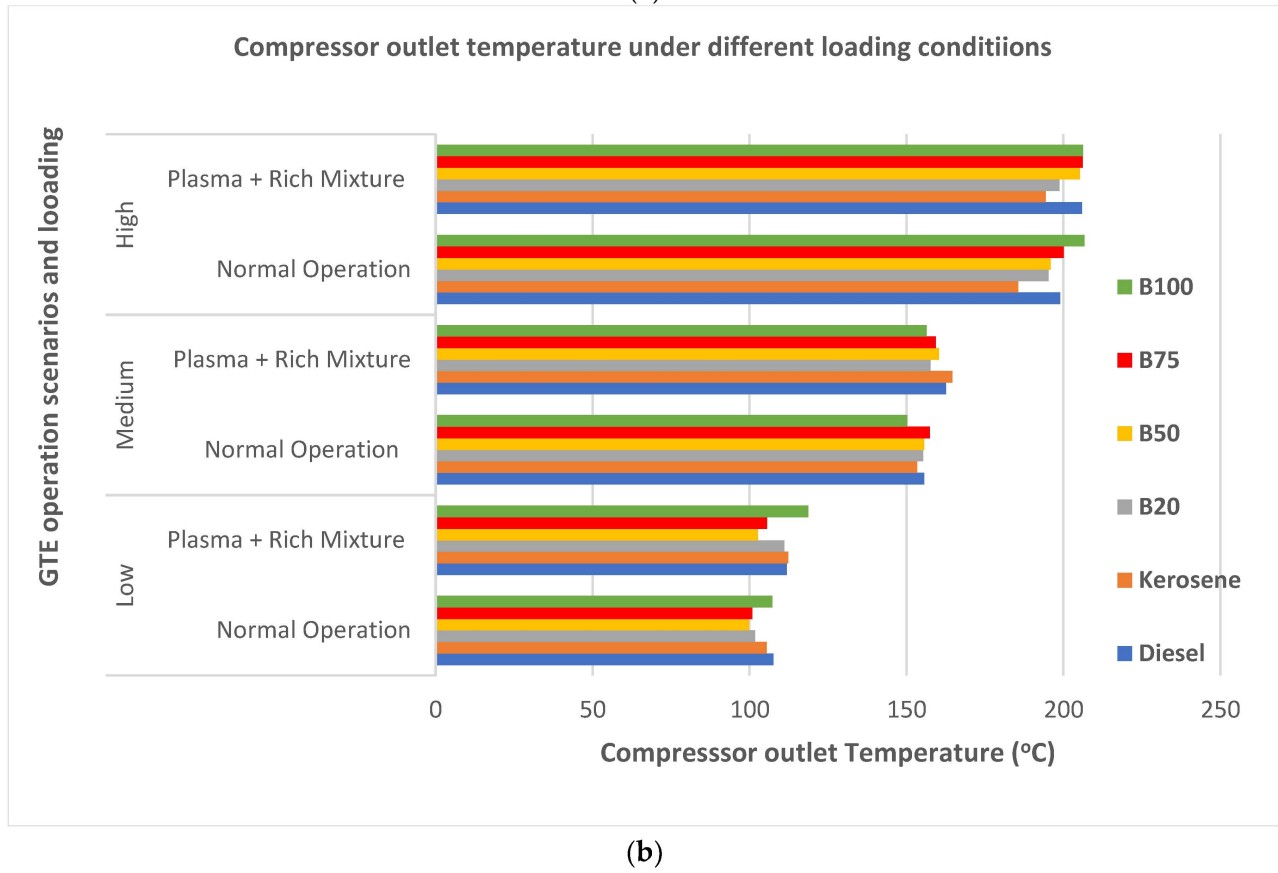

(**b**)

**Figure 8.** (**a**) Compressor inlet temperature comparison plot. (**b**) Compressor outlet temperature comparison plot.

The integrated hybrid plasma-rich fuel mixture technology also increases the compressor outlet temperature of the MGT with increased loading, as displayed in Table 4. The highest compressor outlet temperature in the fabricated GTE was recorded under integrated hybrid plasma-rich fuel mixtures B75 and B100 under low and high loading conditions, as displayed in Table 5. A temperature incremental percentage value of 100.6% for B75 and 87.6% for B100 was observed across the three loading states of the GTE for the six deployed fuel results highlighted in Table 5. The highest compressor outlet temperature of 206 °C was recorded under the highest loading condition by adopting a B75 blend mixture. The integrated hybrid plasma-rich fuel mixture technology enhanced the engine's combustion performance more than under fossil fuel operation, as displayed in Figure 8b. The result implies a positive effect of the introduced plasma technology in increasing the internal heating temperature of the produced hydrogen plasma in the GTE, supporting effective fuel combustion within the combustion chamber.

**Table 5.** Compressor outlet temperature under different loading conditions.

| Mini GTE Loading | MGT Engine Operation Scenarios | Compressor Outlet Temperature Across Adopted Fuel Samples (°C) | | | | | |
|---|---|---|---|---|---|---|---|
| | | Diesel | Kerosene | B20 | B50 | B75 | B100 |
| Low | Normal Operation | 107.6 | 105.5 | 101.7 | 99.9 | 100.9 | 107.3 |
| | Plasma + Rich Mixture | 111.9 | 112.3 | 111.1 | 102.7 | 105.6 | 118.7 |
| Medium | Normal Operation | 155.6 | 153.4 | 155.3 | 155.6 | 157.4 | 150.3 |
| | Plasma + Rich Mixture | 162.6 | 164.6 | 157.7 | 160.4 | 159.3 | 156.4 |
| High | Normal Operation | 199 | 185.6 | 195.3 | 195.9 | 200.1 | 206.7 |
| | Plasma + Rich Mixture | 205.9 | 194.4 | 198.7 | 205.2 | 206.2 | 206.3 |

In the Table 5, yellow highlights indicate the biodiesel improved value from integrated plasma and rich fuel mixture.

### 3.3. Compressor Outlet Pressure

The operational pressures of the fabricated MGT engine, compressor, and auxiliary devices were determined using 5 VDC pressure gauges with specialized sensors made from a carbon steel alloy, with operation ranges between 0 and 2.4 MPa. Other pressures monitored included: pressure before compression, pressure after compression, fuel system pressure across all fuels (kerosene, diesel, and biodiesel), and engine oil system pressure. A hot-wire anemometer (AR866A hot wire anemometer) was used to measure the engine's air mass flow rate to illustrate the effect of plasma-assisted combustion technology on the MGT engine with a fixed fuel flow rate under varying loading conditions. The displayed result indicated constant flow rate values of 360, 460, and 565 mL/min for low, medium, and maximum loading operation conditions, respectively, for the fabricated MGT engine. The obtained output pressure result demonstrated a higher value under an external integrated plasma-rich fuel mixture than under normal working conditions and varied loading conditions (Table 6). The integrated plasma technology increases the compressor outlet pressure with increased in GTE loading across all applied fuels, as highlighted in green. Under normal and integrated plasma technology, the combustion-assisted kerosene presented the most elevated pressure: 11.83 (psi) and 12.02 (psi) under low loading conditions.

**Table 6.** GTE compressor outlet pressure.

| Operation Scenarios | GTE Compressor Outlet Pressure under Constant Fuel Flow Rate (Psi) Values of 360, 460, and 565 mL/min For Low, Medium, and Maximum Loading Operation Conditions | | | | | | |
|---|---|---|---|---|---|---|---|
| | Loading Conditions | Diesel | Kerosene | B20 | B50 | B75 | B100 |
| Normal Operation | Low | 10.28 | 11.83 | 10.85 | 10.77 | 10.68 | 10.91 |
| | Medium | 20.02 | 21.06 | 20.9 | 21 | 21.38 | 19.68 |
| | High | 30.36 | 29.13 | 30.46 | 30.94 | 31.38 | 31.71 |
| Integrated Plasma-Rich | Low | 10.55 | 12.02 | 10.91 | 10.53 | 11.05 | 11.41 |
| | Medium | 20.87 | 22.43 | 21.94 | 21.45 | 22.48 | 19.62 |
| | High | 30.58 | 29.37 | 30.32 | 31.13 | 32.2 | 31.07 |

In the Table 6, yellow and green highlights indicate the comparison result under regular and integrated plasma-rich fuel mixtures.

In contrast, under medium and high loading conditions, B75 presented the most elevated pressure of 21.48 (psi) and 32.2 (psi), respectively, under integrated plasma technology, as highlighted in green in Table 6. These values are higher than those obtained with other fossil fuels, as displayed in Figure 9. It can be deduced that all biodiesel fuel blends demonstrated exemplary performance in terms of wall pressure increase under medium and high loading conditions of the MGT engine at a low fuel injection flow rate (FFR). During this period, the B75 biodiesel blend from animal fat exhibited the best performance in terms of pressure increase, with a marginal flow rate compared to other fuel types.

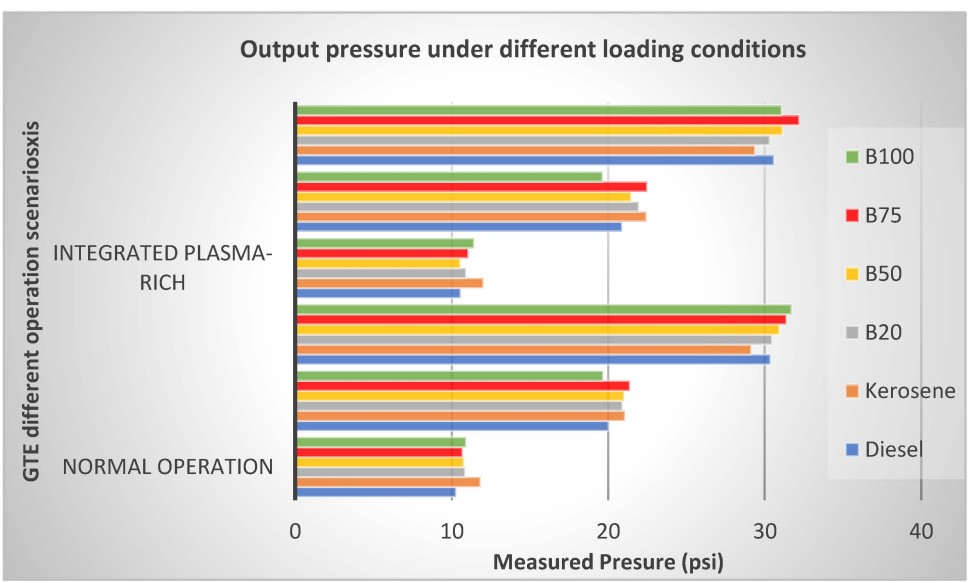

**Figure 9.** Compressor output pressure comparison plot.

It can be deduced that all fuels demonstrated exemplary performance in terms of wall pressure increase at a low fuel injection flow rate. However, at higher injection FFR, the pure biodiesel (B100) exhibited the best performance in terms of pressure increase, with a marginal flow rate compared to other fuel types. All the fuels demonstrated a pressure increase increased injection FFR.

### 3.4. Thrust-Specific Fuel Consumption (TSFC)

The thrust-specific fuel consumption (TSFC) represents the fuel efficiency of an engine design with consideration of the thrust output. The TSFC may also be seen as the fuel consumption (grams/second) per unit of thrust kilonewtons (kN). This means that the fuel consumption is divided by the thrust. TSFC is the mass of fuel needed to provide the net thrust for a given period from an engine. The thrust value is a function of the engine loading that also impacts the fuel consumption level for each applied fuel result (Table 7). Load

cells are transducer instruments used to generate an electrical signal proportional to the load or force applied to the engine. The thrust load cell (OPTIMA, OP-312) is very accurate and rugged and constructed to last a lifetime. This single-load cell system can withstand a load of more than 900 kg within the acceptable load applied in this study. It is constructed with an S-beam made from nickel-plated alloy steel. A weight scale (Vacuum gauge 69086, Yellowjacket, Kyburz, CA, USA) device was used to calculate the fuel consumption rate of the fabricated MGT engine measure the fuel entering the system.

**Table 7.** Thrust value per loading function.

| Fuel Types | Loading Operating Condition (kgf) | | | | | |
| | Normal Operating Scenario | | | External Integrated Plasma-Rich Fuel Mixture Scenario | | |
| | Low | Medium | High | Low | Medium | High |
| Diesel | 1.7 | 2.2 | 4.1 | 1.8 | 2.45 | 4.3 |
| Kerosene | 1.81 | 2.3 | 4.1 | 1.9 | 2.5 | 4.4 |
| B20 | 1.7 | 2.1 | 4.2 | 1.82 | 2.3 | 4.3 |
| B50 | 1.65 | 2.1 | 4.2 | 1.38 | 2.3 | 4.3 |
| B75 | 1.65 | 2.1 | 4.1 | 1.74 | 2.3 | 4.35 |
| 100 | 1.65 | 2.2 | 4 | 1.8 | 2.4 | 4.2 |

In the Table 7, green highlights indicate the loading comparison result under regular and integrated plasma-rich fuel mixtures.

The proper thrust is much higher under integrated plasma-rich fuel mixture atomization. Under low loading, B20 presented the highest thrust value of 1.82 kgf, and the B75 blended mixture presented a value of 4.45 kgf under the highest engine loading conditions using the same fuel volume. The impact of plasma technology is reflected in both fossil fuels under low and medium loading conditions, with much higher values than those obtained with biodiesel blends. The obtained TSFC for B75 under the proposed integrated plasma technology is increased across all loading conditions, with the highest recorded value of 4.35 kgf approximately equal to that obtained from the fossil kerosene under the maximum loading condition, as seen in Table 6. The obtained thrust from an integrated hybrid plasma-rich fuel mixture is enhanced for all the fuels used compared to regular engine operation without plasma technology.

*3.5. Emission Levels of the GTE*

A gas analyzer determines the percentage composition of exhaust gas emissions (CO, NOx, CO2, etc.), differential flue temperature, and ambient carbon monoxide compositions. The influence of the load variation impacts the MGT engine fuel combustion rate under operation with an integrated hybrid plasma-rich fuel mixture. The exhaust released for all loading conditions is displayed in Table 8.

The concentrations across all measured GHG emissions from the fabricated GTE under normal and integrated plasma technology demonstrated the lowest value with the biodiesel fuels. This is due to low values of unburnt hydrocarbon and nitrogen per million liters of applied fuel in the combustion chamber of the GTE. The significant emissions of the MGT are oxygen ($O_2$), carbon monoxide (CO), carbon dioxide ($CO_2$), nitrogen monoxide (NO), and nitrogen dioxide ($NO_2$). The released $O_2$ emissions under plasma-assisted combustion technology do not depend on the fuel type and engine loading. CO is mainly formed through incomplete combustion of fuel within the engine combustion chamber due to insufficient available oxygen. There is variation in CO emissions from different fuels under varying MGT engine loading conditions, with B75 recording 0.07% and 2.8% content values for CO and $CO_2$, respectively, as displayed in Figure 10.

**Table 8.** Measured GTE emission contents in percentage.

| Fuel Types | Load Types | O$_2$ (%) | | CO (%) | | CO$_2$ (%) | | NO (%) | | NO$_2$ (%) | |
|---|---|---|---|---|---|---|---|---|---|---|---|
| | | Norm | Plasma | Norm | Plasma | Norm | Plasma | Norm | Plasma | Norm | Plasma |
| Diesel | Low | 16.4 | 16.3 | 0.17 | 0.18 | 3 | 3.1 | 11 | 12 | 10 | 10 |
| | Medium | 16.7 | 16.7 | 0.1 | 0.1 | 2.8 | 2.9 | 13 | 12 | 13 | 14 |
| | High | 16.6 | 16.6 | 0.07 | 0.08 | 2.9 | 3 | 15 | 15 | 14 | 17 |
| Kerosene | Low | 16.5 | 16.5 | 0.11 | 0.12 | 2.9 | 2.9 | 12 | 11 | 5 | 17 |
| | Medium | 16.7 | 16.6 | 0.08 | 0.09 | 2.7 | 2.8 | 10 | 12 | 10 | 12 |
| | High | 16.8 | 16.5 | 0.05 | 0.06 | 2.7 | 2.9 | 16 | 16 | 4 | 12 |
| B20 | Low | 16.5 | 16.4 | 0.12 | 0.13 | 2.9 | 3 | 12 | 6 | 5 | 10 |
| | Medium | 16.7 | `16.09` | 0.09 | 0.1 | 2.8 | 2.8 | 8 | 7 | 12 | 12 |
| | High | 16.6 | 16.6 | 0.04 | 0.04 | 2.9 | 2.9 | 14 | 15 | 12 | 15 |
| B50 | Low | 16.5 | 16.3 | 0.13 | 0.14 | 2.9 | 3.1 | 9 | 10 | 7 | 9 |
| | Medium | 16.7 | 16.7 | 0.11 | 0.12 | 2.8 | `2.8` | 7 | 7 | 13 | 13 |
| | High | 16.7 | 16.5 | 0.05 | 0.06 | 2.9 | 2.9 | 16 | 16 | 12 | 16 |
| B75 | Low | 16.6 | 16.6 | 0.13 | 0.13 | 2.9 | 2.9 | 6 | 5 | 8 | `9` |
| | Medium | 16.8 | 16.8 | 0.12 | 0.13 | 2.8 | `2.8` | 6 | `5` | 13 | 13 |
| | High | 16.7 | 16.6 | 0.06 | `0.07` | 2.9 | 3 | 16 | 13 | 12 | 14 |
| B100 | Low | 16.4 | 16.4 | 0.18 | 0.18 | 3 | 3.1 | 7 | 8 | 10 | 11 |
| | Medium | 16.8 | 16.7 | 0.13 | 0.13 | 2.8 | 2.9 | 7 | 6 | 14 | 14 |
| | High | 16.6 | 16.6 | 0.08 | 0.08 | 3 | 3 | 12 | 12 | 15 | 15 |

In the Table 8, yellow highlights indicate the GHG emissions comparison result under regular and integrated plasma-rich fuel mixtures.

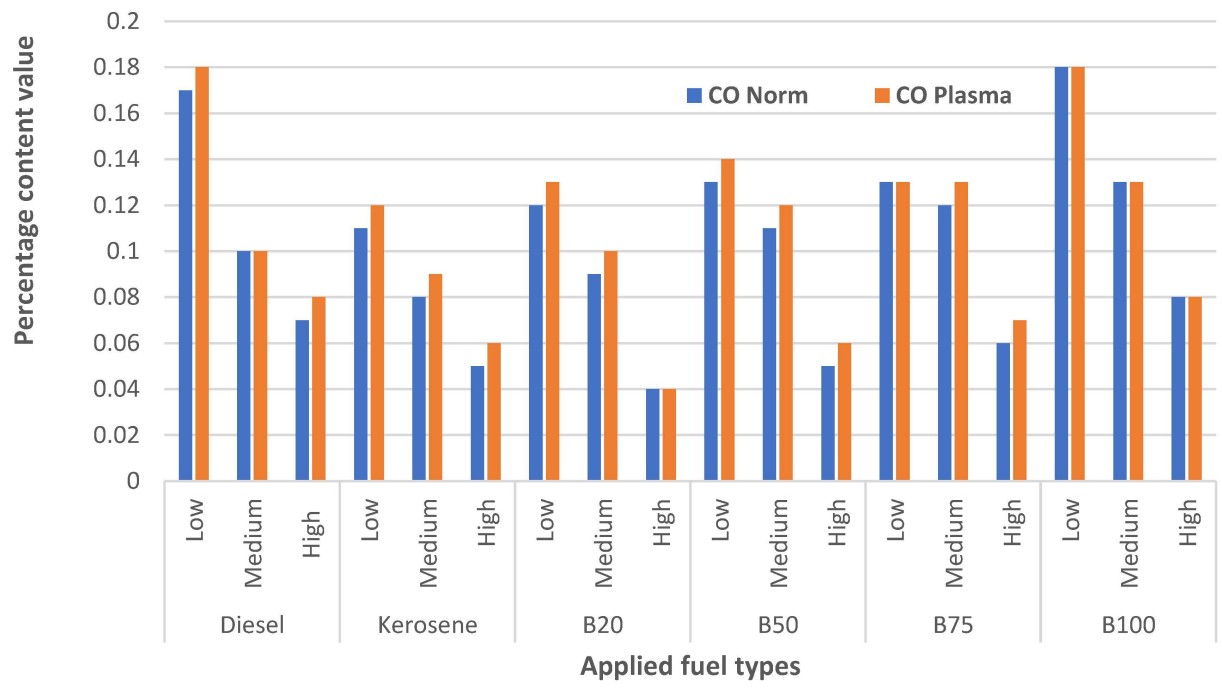

**Figure 10.** CO and CO$_2$ emission contents comparison plot.

The oxides of nitrogen in the exhaust emissions contain nitrogen monoxide (NO) and nitrogen dioxide (NO$_2$), which are influenced by the fuel's cylinder pressure, temperature, and oxygen content. The adoption of plasma technology in the fabricated GTE reduced the

impact of NO and NO$_2$ on the environment. All of the biodiesel fuels produced a promising lower composition of NO emissions under both low and high loading conditions, as seen in the highlighted cells in Table 7. The B75 biodiesel blended fuel demonstrated the lowest percentage of NO and NO$_2$ content of 5% and 7%, respectively, compared with other applied fuels under three different loading conditions, as displayed in Figure 11. Under integrated plasma technology, the obtained NO from with B75 demonstrated lower values under different loading conditions than fossil diesel.

**GTE NO and NO2 emission level coomparison plots**

**Figure 11.** NO and NO$_2$ emission comparison plot.

The lower NO composition is observed under integrated hybrid plasma-rich fuel-assisted combustion, which is attributed to the positive impact of the generated hydrogen plasma molecules that increase the combustion temperature and ensure a complete combustion process. This phenomenon raises both the cylinder temperature and the hydrogen flame formation speed, thus reducing NO emissions from the applied biodiesel fuel blends. This supports the previously obtained result of low sulfur contents from the laboratory fuel characterization tests, as shown in Figure 12. These values increase with engine loading conditions. B20 and B75 resulted in the most reduced emissions, with the lowest values for nitrogen oxide (NO) compared with fossil kerosene, fossil diesel, and other biodiesel blends. Minimizing nitrogen oxide emissions will significantly improve the ozone layer. These results show that integrated hybrid plasma-rich fuel-mixture-assisted technology in engine operation with the use of biodiesel fuels results in a lower composition of NO$_2$ compared to diesel and kerosene.

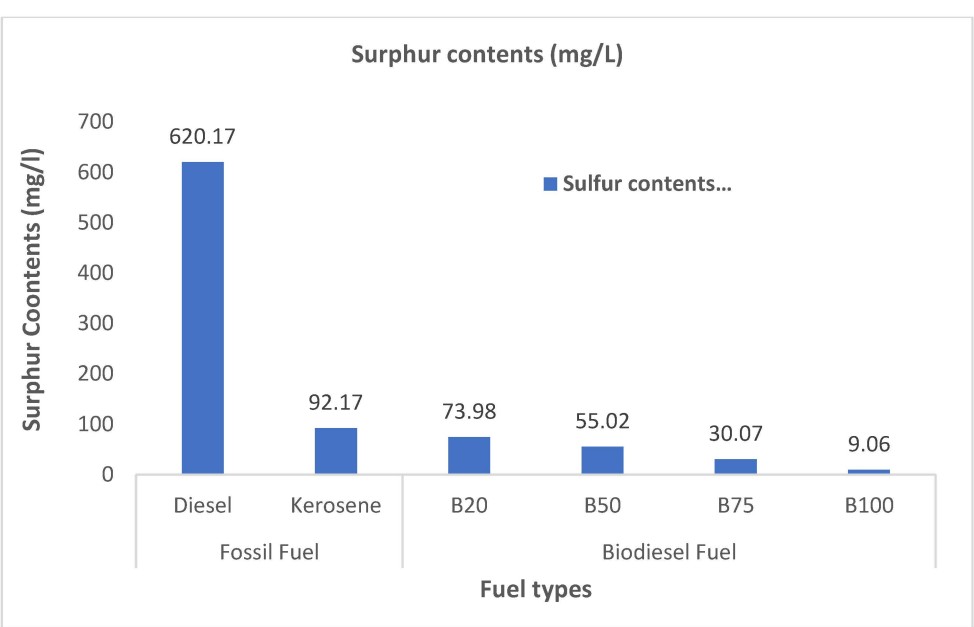

**Figure 12.** Sulfur content comparison plot.

### 4. Conclusions

The B75 blended fuel satisfied most of the stated objectives. The necessary fuel properties, such as a flashpoint value of 62 °C, were closest to those of fossil kerosene. These results demonstrate the potential application of biodiesel products from sheep fats as bio-renewable organic fuel to operate the assembled 50 kW (67 hp) MGT engine in the laboratory. The high lubricity and viscosity of the biodiesel blended mixture assisted in reducing engine wear, achieving a high-pressure increment of 0.5 (psi) and improved GTE power efficiency. The blended biodiesel fuels produced minimal soot during combustion for better engine performance and a reduction in fuel consumption by 9%.

GTE operation under integrated plasma at the compressor inlet positively impacted the system performance efficiency, with increases in internal and external combustion temperatures of 13.3 °C and 8.1 °C, respectively. The B100 and B75 biodiesel blended fuels improved the combustion of hydrocarbons and toxic nitrogen gas content, with low emissions of CO, $CO_2$, NO, and $NO_2$ across all biodiesel blended fuels compared to the fossil kerosene and diesel fuels. The achieved GHG emissions were, on average, 0.07% for CO, 3% for $CO_2$, 5% for NO, and 10% for $NO_2$. The engine test obtained the lowest value of sulfur, carbon, and nitrogen contents, which was a better performance than that of fossil diesel.

The thrust value under standard conditions is 1.7–4.2 kgf, compared to 1.8–4.35 under an integrated plasma system. The obtained average thermal efficiency is between 15 and 18% for biodiesels. In this study, we achieved a set objectives for the potential of biodiesel as an alternative renewable fuel source for gas turbine operation. Future studies should conduct tests on the durability of engine parts for practical biodiesel usage.

The following are the contributions of this work to the existing body of knowledge.

i.   Successful fabrication and assembly of a laboratory-scale MGT engine with external integrated plasma torch technology and an ultrasonic atomizer at the compressor inlet of the testbed MGT engine.
ii.  Comparative reduction in the GHG emissions of $NO_2$, $SO_2$, and CO achieved by introducing plasma combustion technology for the MGT engine using biodiesel fuel as opposed to conventional approaches.
iii. Performance improvement of MGT at a reduced operational cost.

**Author Contributions:** Conceptualization, A.F.; Resources, A.A. and A.N.; Supervision, N.M.A., M.K.A.B.M.A. and H.A.A.; Writing—original draft, A.M.R.N.A. All authors have read and agreed to the published version of the manuscript.

**Funding:** This research received no external funding. Self founding (Ahmed M. R. N. Alrashidi).

**Institutional Review Board Statement:** Not applicable.

**Informed Consent Statement:** Not applicable.

**Data Availability Statement:** Experimental data with the main author.

**Conflicts of Interest:** The authors declare no conflict of interest.

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
