# Peer review of "Impact of Plasma Combustion Technology on Micro Gas Turbines Using Biodiesel Fuels"

_applsci, doi:10.3390/app12094321_

Round 1

Reviewer 1 Report

The work is well done and deserve journal publication.

I just have one question for the authors regarding the global feasibility of using biofuels, would it be possible to almost completely replace the use of fossil fuels with biofuels? May you argue about this in the paper?

Author Response

Reviewer's comment and corrections

S/N

Comments

Corrections

Page/line number

1

Does the introduction provide sufficient background and include all relevant references?

Improved on the background and references

Pg  1-3

2

I just have one question for the authors regarding the global feasibility of using biofuels, would it be possible to almost completely replace the use of fossil fuels with biofuels? May you argue about this in the paper?

It may be sustainable if global waste is harnessed for this purpose. Considering the large volume of such waste generated daily across the globe. Biodiesel from plant-based oils gives competition for food but biodiesel from animal fat wastes is cost-effective whilst reducing its impact on the environment. Also, integration of cost-effective Kuwaiti sheep fat biodiesel and plasma technology in the operating of GTEs is lacking.

Line 94 -97

Reviewer 2 Report

The paper is interesting, well written and easy to follow based on experimental tests. Few comments can be considered as follows:

If Fig 3a is the one used, if not could you please replace it with the existing one?

You can write the format of figures and tables according to the template of the journal.

Fig 8b Compresure to be replaced by compressor.

You write a note under the tables describing the values that are highlighted in yellow and green.

Conclusion requires more details and the authors can point out the main findings from the paper.

The authors need to write the main specifications of the engine.

The authors can add a section of nomenclature.

The authors can describe "Micro Gas Turbine Engine" for non-specialist readers.

What is the value of loading conditions in kW?

What is the value of fuel consumption in g/kw.h or g/h?

The authors can follow the refs guidelines of the journal.

The authors can comment about using ultra-low sulphur fuel (diesel)? is there any effect can occur other than the normal one used?

Author Response

Reviwers comment and coorrectins

S/N

Comments

Corrections

Page/line number

1

If Fig 3a is the one used, if not could you please replace it with the existing one?

The figure 3a is replaced with the right one used for the research

Pg  5

2

You can write the format of figures and tables according to the template of the journal.

The figures and tables format complied with the journal stated format

All pages

3

Fig 8b Compresure to be replaced by compressor.

Grammatical errors attended to across the manuscript

Pg  13

4

You write a note under the tables describing the values that are highlighted in yellow and green.

Notes written on each Table highlight and explanations provided in the text

All pages

5

Conclusion requires more details and the authors can point out the main findings from the paper.

Conclusion section improved upon with details provided

Pg  18 -19

6

The authors need to write the main specifications of the engine.

Engine specifications provided within the text

Table 1 in Pg. 4

7

The authors can add a section of nomenclature.

Engine nomenclature added

Table 1 in Pg. 4

8

The authors can describe "Micro Gas Turbine Engine" for non-specialist readers.

The brief description of the MGT engine provided within the text

Lines 44 - 66

9

What is the value of loading conditions in kW?

The engine loading capacity is 50 kW (67 hp)

Line 126

10

What is the value of fuel consumption in g/kw.h or g/h?

The consumption varied with different applied fuel types.  a weight scale (Vacuum gauge 69086, Yellow jacket, USA) device was used to calculate the fuel consumption from the engine, and also record the amount of fuel entering the system

Pg  15

11

The authors can follow the ref's guidelines of the journal.

The referencing style followed

Pg  20 -22

12

The authors can comment about using ultra-low sulphur fuel (diesel)? is there any effect can occur other than the normal one used?

Not considered because it is out of scope for this study. It may be considered in future studies,

Reviewer 3 Report

  • This is generally an interesting topic and the paper makes a useful contribution to the literature although, personally, I found the employment of animal products in this application distasteful.
  • The level of English language of the paper is not very good. The first sentence supports my argument, “The biofuel adoption in the internal combustion engine of automobile and gas turbine engines (GTE) is a generally acceptable phenomenon”. A generally acceptable phenomenon is a vague expression lacking in academic rigour.
  • In the second sentence the authors write, ... with the conventional use of fossil fuel resources in the transportation sector. Is the use that is conventional or is it the case that what is conventional is the fossil fuel? Therefore I think that it would be better if the article was proofread by a native English speaking person. As a suggestion, based on the experience of some of my students, I will direct the authors to this site, https://www.leproofreading.com

Technically the paper is not particularly sound although it has a good structure and a well presented set of results. My objections stem from the fact that many important information is not adequately explained and often missing entirely.

How were the temperatures measured, which sensors, where, were the measurements repeatable, etc? Same for other quantities which are not clearly discussed either. This is critical information and therefore must be included in the final version of this article.

Figure 2 - plasma touch instead of plasma torch

Line 149 - pressure ratio 2:6? Should read 2.6?

What is the source of the compressor map?

Figure 3b is grossly distorted in terms of its actual real dimensions. Presumably this was done in order for the two parts of Figure 3 to have the same height but distorted images aren’t acceptable for inclusion in scientific articles. The same reasoning is applied to Figure 6, particularly 6a which looks ridiculously distorted.

Line 159 - 2.2.2 The exhaust nozzles redesigned. How many exhaust nozzles are there? Redesigned? This section should be clearer about the actual characteristics of the nozzle including quantitative information.

Table 2/3/... what is the purpose of the coloured cells, what do they indicate?

Section 2.4 The Micro GTE operation, Testing, and Measurements deals with the instrumentation and testing. What did it say? That key parameters are ... monitored using different sensors at different locations ... This level of information is totally worthless in a research paper. The authors need to please find a way of describing what was measured, where, how. The description can be succinct but it has to be included.

Three load conditions are mentioned. How are these characterised, what parameters are employed in their definition?

Regarding the temperatures, how are these related to the varying atmospheric conditions?

Likewise the pressures, Figure 9. That’s why it would be expected to have these quantities non-dimensionalised.

Please check carefully your spelling. The caption of Figure 8 has got three words misspelled.

Are the conclusions directly related to the measured parameters, ie efficiency observations?

It is suggested that the authors need to review this article thoroughly before it reaches an acceptable publication standard.

Author Response

Reviewer's comment and corrections

S/N

Comments

Corrections

Page/line number

1

Is the research design appropriate?

The research design improved upon

Pg 3 -10

2

Are the methods adequately described?

Methodology improved upon to reflect details for each construction step and standards

Pg 3 -10

3

Are the results clearly presented?

All result is presented with supporting values

Pg 10 – 18

4

Are the conclusions supported by the results?

The conclusion reflects the implication of the discussed result for further supports

Pg 18– 19

5

The level of English language of the paper is not very good. The first sentence supports my argument, “The biofuel adoption in the internal combustion engine of automobile and gas turbine engines (GTE) is a generally acceptable phenomenon”. A generally acceptable phenomenon is a vague expression lacking in academic rigour.

In the second sentence the authors write, ... with the conventional use of fossil fuel resources in the transportation sector. Is the use that is conventional or is it the case that what is conventional is fossil fuel? Therefore I think that it would be better if the article was proofread by a native English speaking person. As a suggestion, based on the experience of some of my students, I will direct the authors to this site, https://www.leproofreading.com

Manuscript subjected to proofreading using Grammarly English editor

All pages

6

Technically the paper is not particularly sound although it has a good structure and a well-presented set of results. My objections stem from the fact that many important information is not adequately explained and often missing entirely.

How were the temperatures measured, which sensors, where, were the measurements repeatable, etc? Same for other quantities which are not clearly discussed either. This is critical information and therefore must be included in the final version of this article.

Details across all steps adopted and the instrument used for each parameter measurement are well explained in the revised version of the manuscript with attention to details

Pg 11 for temperature tester, Pg. 14 for pressure tester Pg. 14 for the mass flow rate entering the engine, and Pg 16 for GHG emission contents

7

Figure 2 - plasma touch instead of plasma torch

Grammatical error corrected

Pg 5

8

Line 149 - pressure ratio 2:6? Should read 2.6?

Pressure ratio corrected

Lines 196

9

What is the source of the compressor map?

10

Figure 3b is grossly distorted in terms of its actual real dimensions. Presumably this was done in order for the two parts of Figure 3 to have the same height but distorted images aren’t acceptable for inclusion in scientific articles. The same reasoning is applied to Figure 6, particularly 6a which looks ridiculously distorted.

Figure 3b and Figure 6a are replaced and resized appropriately to an acceptable standard

Pg 5 and 7 respectively

11

Line 159 - 2.2.2 The exhaust nozzles redesigned. How many exhaust nozzles are there? Redesigned? This section should be clearer about the actual characteristics of the nozzle including quantitative information.

There are four units of single injector nozzles attached to the inlet manifold.

Pg 6. Line 209 - 210

12

Table 2/3/... what is the purpose of the coloured cells, what do they indicate?

The purpose for the highlight explained within the text and the small attached note under each table

Result and discussion section

13

Section 2.4 The Micro GTE operation, Testing, and Measurements deals with the instrumentation and testing. What did it say? That key parameters are ... monitored using different sensors at different locations ... This level of information is totally worthless in a research paper. The authors need to please find a way of describing what was measured, where, how. The description can be succinct but it has to be included.

All parameters measured and instruments deployed for the measurement are explained within the text for more clarification.

Pg 11 for temperature tester, Pg. 14 for pressure tester Pg. 14 for the mass flow rate entering the engine, and Pg 16 for GHG emission contents

14

Please check carefully your spelling. The caption of Figure 8 has got three words misspelled.

All grammatical errors attended to using the English editing tool

15

Are the conclusions directly related to the measured parameters, ie efficiency observations?

The conclusion is directly related to the results discussed. The direct implication of all results

Pg. 18 - 19

16

It is suggested that the authors need to review this article thoroughly before it reaches an acceptable publication standard.

The manuscript has been improved upon both in semantic and syntax

All pages

Round 2

Reviewer 3 Report

The authors have introduced a number of revisions to the article and the outcome has led to a moderate improvement to the quality of the paper.

My main concern is still not addressed however, namely the standard of the English language.

One such example is the following sentence:

“Minimizing nitrogen oxides emissions will greatly improve the ozone layer, and from these results, it is obvious that integrated hybrid plasma-rich fuel mixture assisted technology in the engine operation with the use of biodiesel fuels emit lower composition of NOx compared to diesel and kerosene with emission level lower.”

This sentence has a vague conclusion meaning.

Lower composition of NOx? What is the meaning of, lower composition of NOx?

What is then … emission level lower …. referring to, NOx again?

Unfortunately the paper is still full of similar bad grammar examples.

The article needs to be proofread.

A small amount of English transgressions is to be expected from non native English speakers but not when the volume is such that it dominates the whole article.

Another example. “The increased inlet temperature is to support the combustion chamber temperature build-up for effective combustion performance of the engine for continued heating achievement”.

What is the meaning of the expression, continued heating achievement? And effective combustion performance?

Etc.

The article needs to be proofread.

Elsewhere I have still got a number of objections namely the reporting of the experimental procedure. The authors are directed to a paper called “Experimental Analysis of a Micro Gas Turbine Fuelled with Vegetable Oils from Energy Crops” by Cavarzere et al specifically the Section 4 detailing the experiments. Nobody wants to know what part number! the device had, what the reader would like to know is the type of sensor, and particularly its accuracy. That is a basic requirement that the suggested paper addressed. To say that the load cell is very accurate is not acceptable. We need values as is done by the Italian article.

Figure 11 has four legends but only two colours. Am I missing something?

In conclusion, the paper requires additional work to bring it to a publishable standard.

Author Response

Reviewer 3 comment and corrections

S/N

Comments

Corrections

Page/line number

1

My main concern is still not addressed however, namely the standard of the English language.

The manuscript's English language style and structure have been improved upon after sending the paper for professional proofreading

All pages

2

Figure 11 has four legends but only two colours. Am I missing something?

The additional legends has been removed from figure 11

Pg 18

3

In conclusion, the paper requires additional work to bring it to a publishable standard.

Manuscript comprehensively improved upon
